# There is More to Graphs than Meets the Eye: Learning Universal Features with Self-supervision

## Abstract

We study the problem of learning universal features from multiple graphs through self-supervision. Graph self-supervised learning has been shown to facilitate representation learning, and produce competitive models compared to supervised baselines. However, existing methods of self-supervision learn features from one graph, and thus, produce models that are specialized to a particular graph. We hypothesize that leveraging multiple graphs of a family can improve the quality of learnt representations in the model by extracting features that are universal to the family of graphs. However, learning universal features from disparate node/edge features in different graphs is challenging. To address this challenge, we first homogenise the disparate features with graph-specific modules that feed into a universal representation learning module for generalisable feature learning. We show that leveraging multiple graphs of the same family improves the quality of representations and results in better performance on downstream node classification task compared to self-supervision with one graph. In this paper, we present a principled way to design foundation graph models that are capable of learning from a set of graphs in a holistic manner. This approach bridges the gap between self-supervised and supervised performance, while reducing the computational time for self-supervision and parameters of the model.

## 1 Introduction

Graphs are rich and expressive mathematical abstractions and data structures that can represent properties of nodes and links in interconnected systems such as social networks, molecules and knowledge graphs. Several graph neural networks (GNNs) and graph transformers (GTs) that leverage powerful data processing architectures have been proposed to learn challenging tasks for a wide range of datasets (Fan et al., 2019; Baek et al., 2020; Rong et al., 2020; Muzio et al., 2021; Wu et al., 2022). Recently, self-supervised learning (SSL), which was born out of natural language processing and was later successfully applied to computer vision (Kolesnikov et al., 2019; He et al., 2022), has been demonstrated to aid in graph representation learning. SSL has ushered in more powerful GNN and GT models (Rong et al., 2020; Xiao et al., 2022; Jin et al., 2022; Liu et al., 2022). The success of self supervised models for graph-structured data has been demonstrated in different applications such as recommendation systems (Wu et al., 2021) and molecular property prediction (Zhang et al., 2021).

Self-supervision exploits unlabelled data to learn representations that might be useful across many downstream tasks (Balestriero et al., 2023). The state-of-the-art (SOTA) in graph self-supervision constrains pre-training to only one dataset (e.g., `CoraFull`), with one (Liu et al., 2022) or many (Jin et al., 2022) pre-training tasks at a time. As a result, representations learnt through SSL are likely to be specialized to one particular dataset, and thus lack the ability to generalize well to other graphs (e.g., `DBLP`) of the same family (here, citation networks). Thus, state-of-the-art graph SSL entails individualized pre-training for every dataset of interest, and exhibits several drawbacks. First, each model learns a distinct set of parameters, independent of other similar datasets. Such a model does not leverage any shared parameters that could lead to learning universal features, nor does it exhibit the ability to exploit data from other datasets during training. This hampers generalizability of the resulting models, and as shown in this work, also the performance of SSL models on downstream

node classification tasks. Second, owing to the different node and edge feature dimensions of different datasets, models obtained with state-of-the-art SSL are not compatible with other datasets. As a result, with the availability of new datasets, it is imperative to build a new model from scratch, and one cannot leverage previously learnt representations to inform the training process and reduce the computational load. In other words, state-of-the-art SSL models do not exhibit adaptability. Finally, training a separate model for each dataset increases the computational cost of self-supervision. Furthermore, multiple models require proportionally more storage, adding to the cost of SSL. On the other hand, graphs belonging to the same family are known to exhibit universal patterns (Sharan et al., 2005; Wang & Barabási, 2021), and consequently, it is important to develop a combined learning framework to simultaneously learn from multiple graphs of a family, leading to a *foundation graph model*.

Learning universal representations across graphs poses an important challenge of disparate node and edge features for different graphs. Specifically, node features of different graphs typically exhibit different dimensionality, and do not render themselves to straightforward comparison across graphs. Additionally, the features of different graphs can represent different quantities even if they are of the same dimensionality, hindering unified processing of these features. As a result, it is imperative for a universal SSL approach to be able to accommodate this diversity, and treat disparate node and edge features in a unified manner. Along similar lines, there has been an increased interest in developing models that can handle data of different modalities, and learn features from different sources of data, such as videos and text, through modular structures and carefully crafted embeddings (Gao et al., 2020; Akbari et al., 2021). These foundation multi-modal approaches transform multi-modal data into a common representation space to learn better and robust features. Such an approach has met with incredible success, with a host of different architectures and data processing pipelines developed in the recent years (Lu et al., 2022a;b; Wang et al., 2022; Xu et al., 2023), and is paving the way towards artificial general intelligence (Fei et al., 2022). Inspired by the success of these models, our work aims to investigate if a universal learning approach can be adopted to learn representations from multiple graphs with disparate features, and if the resulting models exhibit better performance in downstream tasks.

**Contributions:** In this work, we propose a generic framework that is rooted in universal representation learning capable of learning universal features from multiple graphs. We use a state-of-the-art graph transformer architecture to construct a universal model, and train it in an end-to-end manner with six benchmark citation networks. We explicitly address the challenges with SSL outlined above, and demonstrate the superiority of the resulting models over traditional approaches. Specifically, this work makes the following contributions:

1. Present a universal representation learning framework through self-supervision with multiple graphs (U-SSL). Our universal model consists of graph-specific parameters that accommodate the disparity of node features of different graphs, and universal parameters that learn representations generic to all graphs used during training. The model can be trained to learn both graph-specific and universal parameters in an end-to-end manner.
2. Present a graph transformer-based U-SSL model, and perform extensive experiments with five benchmark citation network datasets, demonstrating the superiority of the resulting models over those obtained with SSL. We also demonstrate that training universal models is computationally efficient compared to SSL.
3. Demonstrate scalability and adaptability of universal models with a large citation network dataset (`OGBN-arxiv`).

We achieve 1 to 8 points improvement in accuracy on downstream node classification for graphs of different sizes, $6\%$ improvement in training time per epoch, while requiring only a fraction ($0.40$) of the number of parameters compared to SSL. The proposed U-SSL framework is aligned with the core features of foundation models, specifically, it learns across multiple graphs, exhibits properties such as unification, adaptability, and generalizability, and can serve several downstream tasks, resulting in foundation graph models.

## 2 RELATED WORK

**Graph neural networks and graph transformers**   Graph neural networks have been extremely successful in learning representations from graph-structured data, and solving challenging problems in applications including neuroscience (Wein et al., 2021), medicine (Bongini et al., 2021), optimization

(Schuetz et al., 2022) and many more. Most GNN architectures can be broadly categorized as message passing networks, that operate in two stages, i.e., aggregation and combination, with different architectures performing these steps in different ways. One of the earliest GNNs generalized the convolution operation to graph-structured data, and proposed the Graph Convolutional Network (GCN) (Kipf & Welling, 2016). This was followed by an explosion of GNN models, such as GraphSAGE (Hamilton et al., 2017), Graph Attention Networks (GAT) (Veličković et al., 2018) and Graph Isomorphism Networks (GIN) (Xu et al., 2019) that crafted different aggregation and combination operations to capture different relationships in graphs. For instance, GAT uses an attention mechanism for aggregation to assign different weights to different nodes in a neighborhood, allowing the model to focus on the most relevant nodes for a given task, and obtain better performance than GCN that uses convolution for aggregation.

Message passing networks (MPNs) suffer from fundamental limitations, e.g., over-smoothing (Oono & Suzuki, 2020), over-squashing (Alon & Yahav, 2021) and expressive limits (Morris et al., 2019), that are addressed with graph transformers (Rampášek et al., 2022). GTs make use of positional or structural embeddings along with global attention mechanisms to learn both local and global features and thus address the limitations of MPNs (Rampášek et al., 2022). Several GT architectures have been proposed for homogeneous graphs (Yun et al., 2019; Kreuzer et al., 2021), heterogeneous graphs (Hu et al., 2020) and hyper-graphs (Kim et al., 2021). GTs, however, relatively require more training data and do not generalize well to unseen graphs Zhao et al. (2021); Chen et al. (2023b).

**Graph representation learning with self-supervision** SSL learns generic representations as opposed to task-specific representations in supervised learning. There are several SSL methods on graphs including Deep Graph Infomax Velickovic et al. (2019) and Auto-SSL Jin et al. (2022), as well as reviews Jin et al. (2022); Xie et al. (2022); Liu et al. (2022) that summarize the state-of-the-art. Graph SSL has been performed with contrastive as well as predictive learning tasks Xie et al. (2022). While the former aim to learn representations by distinguishing positive and negative samples, the latter seek to predict the values of masked or corrupted nodes or edges. For instance, Velickovic et al. (2019) adopt contrastive learning and maximize mutual information between local patches of a graph and the global graph representation to learn node representation. Rong et al. (2020) apply SSL to molecular graphs to learns representations by predicting masked nodes and edges. There are several SSL tasks such as node attribute masking, graph structure prediction, and graph context prediction, which can be used to learn representations in a self-supervised manner.

The majority of graph self-supervision is performed with one graph and one SSL task. Jin et al. (2022) proposed a mechanism to automate self-supervision with multiple tasks, by adaptively weighing the losses of different tasks during training. Their framework, named Auto-SSL, extended SSL to include multiple tasks during training. However, all SOTA graph SSL methods use only one graph/dataset to learn representations prior to downstream task learning. We address this gap, and a framework to learn universal representations across different graphs – of a certain family.

## 3 LEARNING UNIVERSAL FEATURES WITH GRAPH SELF-SUPERVISION

In this section, we describe the problem formulation and our hypothesis on improving graph representation learning, followed by the construction of different components of the universal self-supervision (U-SSL) model.

### 3.1 PROBLEM FORMULATION AND HYPOTHESIS

We consider $N$ graphs $\{\mathcal{G}_i\}_{i=1}^{N}$, with each graph represented as a tuple of nodes $\mathcal{V}_i$ and edges $\mathcal{E}_i$, $\mathcal{G}_i = (\mathcal{V}_i, \mathcal{E}_i)$ such that $|\mathcal{V}_i| = N_i$ and $\mathcal{E}_i \subseteq \mathcal{V}_i \times \mathcal{V}_i$. Let $\mathbf{A_i} \in \{0,1\}^{N_i \times N_i}$ and $\mathbf{X_i} \in \mathbb{R}^{N_i \times D_i}$ represent the adjacency matrix and node feature matrix of $\mathcal{G}_i$, respectively. Let $\mathcal{L}_{SSL,i}$ denote the pretext task loss for graph $\mathcal{G}_i$. We then provide the definition of SSL, as studied in the current literature as:

**Definition 1.** *For graph $\mathcal{G}_i$, the problem of self supervised learning is to learn an encoder $f_i(\mathbf{X_i}, \mathbf{A_i}; \mathbf{\Theta_i})$ by minimizing the loss $\mathcal{L}_{SSL,i}$ such that the learnt representations can be used to solve downstream learning tasks for $\mathcal{G}_i$.*

We extend this definition to the problem of learning universal features with self-supervision (U-SSL) as follows:

**Definition 2.** *For graphs $\{\mathcal{G}_i\}$, the problem of universal self-supervision is to learn an encoder $f\left(\{\mathbf{X_i}\}, \{\mathbf{A_i}\}; \{\mathbf{\Theta_i}\}, \mathbf{\Phi}\right)$ by minimizing the loss $\sum_{i=1}^{N} \mathcal{L}_{SSL,i}$ such that the learnt features can be used to solve downstream tasks for $\{\mathcal{G}_i\}$.*

The U-SSL model can take as input, disparate features from different graphs, and learn universal features that are common to all the datasets, thereby generalizing well to these datasets, and potentially also to other similar datasets. We note that different graphs have different node feature sizes, i.e., in general, $D_i \neq D_j$ for $i \neq j$. This necessitates that there be parts of the encoder $f$ dedicated to different graphs, with graph-specific parameters $\mathbf{\Theta_i}$, in addition to the universal parameters $\mathbf{\Phi}$.

Let us denote the representations learnt for graph $\mathcal{G}_i$ with SSL as $\mathbf{H_i^s}$, and those learnt with U-SSL as $\mathbf{H_i^u}$, i.e.,

$$\mathbf{H_i^s} = f_i\left(\mathbf{X_i}, \mathbf{A_i}; \mathbf{\Theta_i}\right), \tag{1}$$

$$\mathbf{H_i^u} = f\left(\mathbf{X_i}, \mathbf{A_i}; \mathbf{\Theta_i}, \mathbf{\Phi}\right). \tag{2}$$

Our hypothesis is that U-SSL can learn representations that are better than those learnt with SSL, in terms of solving a downstream task, e.g., node classification, for graphs $\{\mathcal{G}_i\}_{i=1}^{N}$. Let us denote the downstream task head for graph $\mathcal{G}_i$ as $h_i(\cdot; \mathbf{\Psi_i})$, and let $\mathcal{M}$ be a metric such that higher values of $\mathcal{M}$ represent better performing models. Then, our hypothesis can be formally stated as:

$$\mathcal{H}: \quad \mathcal{M}\left(h_i\left(\mathbf{H_i^u}; \mathbf{\Psi_i^u}\right)\right) > \mathcal{M}\left(h_i\left(\mathbf{H_i^s}; \mathbf{\Psi_i^s}\right)\right). \tag{3}$$

Here, the superscripts in $\mathbf{\Psi_i}$ signify that the parameters learnt during fine-tuning of SSL and U-SSL models will be different for the same downstream task head $h_i$.

In formulating our hypothesis, we view a graph $\mathcal{G}_i$ as being an instance of some underlying real-life phenomenon. For instance, `CoraFull`, `DBLP`, `Citeseer`, etc. are different instances of the same underlying real-life phenomenon, i.e., citation among research articles. Learning representations with SSL allows one to extract patterns from only one instance of the underlying phenomenon, while U-SSL allows learning from multiple instances, and hence, observing the underlying phenomenon through multiple lenses. As a result, U-SSL allows learning representations that are fundamental to the underlying mechanism, and is not restricted to the patterns observed in one instance. This can lead to learning more generic features, and hence better downstream performance with U-SSL.

## 3.2 GRAPH-SPECIFIC ENCODER

The core idea of U-SSL is to learn representations that are generalizable across multiple graphs. This entails processing node features from different graphs in a unified pipeline. However, node (and edge) features of different graphs are obtained with different algorithms, and are typically disparate, i.e., (a) they do not have the same dimensionality, and (b) the entries of feature vectors can bear different meanings for different graphs, even if they have the same dimensionality. It is thus imperative to first homogenize the node (and edge) features of different graphs from their original disparate spaces (of dimension $D_i$) to a common space (of dimension $D$) for processing by the rest of the model. We therefore need graph-specific encoders, represented as $g_i(\cdot; \mathbf{\Theta_i})$ for graph $\mathcal{G}_i$. The encoder $g_i$ can be any neural network module, e.g., GCN layers, linear layers, etc. that transforms the feature vectors into $\mathbb{R}^D$, and can additionally involve pre-processing steps such as node feature augmentation to enrich the feature vectors. In our proposed framework, we include feature augmentation ($F_A$) followed by feature transformation ($F_T$), that transform the node features $\mathbf{X_i} \in \mathbb{R}^{N_i \times D_i}$ to $\tilde{\mathbf{X}}_\mathbf{i} \in \mathbb{R}^{N_i \times \tilde{D}_i}$ to $\mathbf{Z_i} \in \mathbb{R}^{N_i \times D}$:

$$\tilde{\mathbf{X}}_\mathbf{i} = F_A\left(\mathbf{X_i}\right), \tag{4}$$

$$g_i(\mathbf{X_i}; \mathbf{\Theta_i}) = \mathbf{Z_i} = F_T\left(\tilde{\mathbf{X}}_\mathbf{i}; \mathbf{\Theta_i}\right), \tag{5}$$

$$= F_T\left(F_A\left(\mathbf{X_i}\right); \mathbf{\Theta_i}\right). \tag{6}$$

In general, the functions $g_i$, $F_A$ and $F_T$ also take the adjacency matrix $\mathbf{A_i}$ as input, which is omitted here for brevity. The output of the graph-specific encoders $\mathbf{Z_i}$ represents the graph-specific homogenized features that exist in $\mathbb{R}^D$, $\forall \mathcal{G}_i$ and whose individual entries represent the same quantity across all graphs. In an $N$-graph application, the U-SSL model will be constructed with $N$ different graph-specific encoders, as shown in Fig. 1.

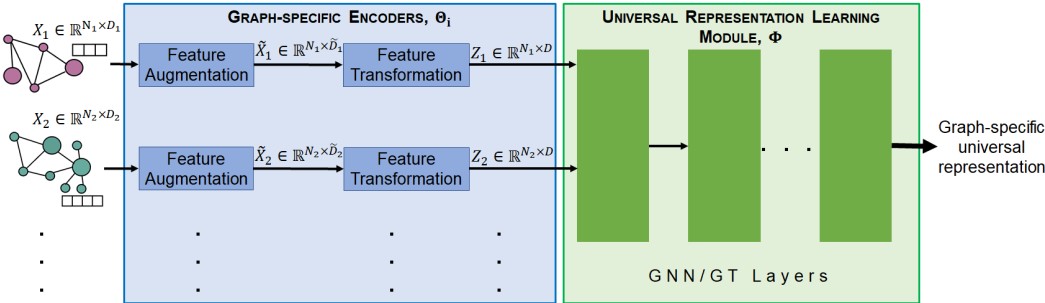

(a) Model architecture for universal self-supervision

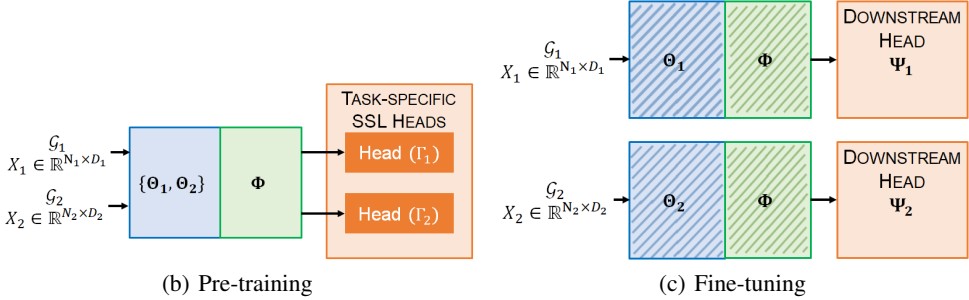

(b) Pre-training                                    (c) Fine-tuning

Figure 1: Universal Self-supervised Learning (U-SSL) across graphs. (a) Model architecture for U-SSL with graph-specific ($\mathbf{\Theta_i}$) and universal ($\mathbf{\Phi}$) parameters. (b) U-SSL pre-training with two graphs, $\mathcal{G}_1$ and $\mathcal{G}_2$ for two tasks with parameters $\mathbf{\Gamma_1}$ and $\mathbf{\Gamma_2}$. (c) Downstream task learning for individual graphs. Hatched boxes represent frozen parameters ($\mathbf{\Theta_i}, \mathbf{\Phi}$), and shaded boxes represent learnable parameters ($\mathbf{\Psi_i}$).

### 3.3 Universal Representation Learning Module

The universal representation learning (URL) module aims to learn features that are generic to all $N$ graphs used during pre-training, and thus capture patterns that are fundamental to the underlying process. It takes in the homogenized node features $\mathbf{Z_i}$ from all graphs, and learns the graph-specific universal features, denoted as $\mathbf{H_i^u}$ for graph $\mathcal{G}_i$. The URL module, denoted as $g(\cdot, \mathbf{\Phi})$ for all graphs $\{\mathcal{G}_i\}$ can be any neural network module, e.g., GNN layers or GT blocks, and can be expressed as:

$$\mathbf{H_i^u} = g(\mathbf{Z_i}; \mathbf{\Phi}), \forall i \in [1, N]. \tag{7}$$

These features are graph-specific since they are obtained from the homogenized node features of a particular graph, and at the same time universal, because they are learnt by minimizing the collective loss accrued for all graphs. A U-SSL model for $N$ graphs is thus constructed with $N$ graph-specific encoders and one universal representation module, as shown in Fig. 1(a). This modular nature of the model architecture allows adding as many graph-specific encoders as desired, and simultaneously processing disparate node features, thus facilitating end-to-end training of the model. In addition, this modular nature renders adaptability to the model, wherein a new graph-specific encoder can be introduced to the model without having to alter the rest of the model structure, and re-train, or continue training with the new dataset.

### 3.4 Pre-training and fine-tuning U-SSL models

Pre-training models with SSL involves selecting one or more pre-training task (also referred to in the literature as pretext tasks), typically depending on the type of downstream task, and appending a model with heads to learn the different tasks. Pre-training of U-SSL models is also performed in a similar vein, i.e., by using the U-SSL model with $N$ graph-specific modules, one universal representation module and one or more task-specific heads. Let $\mathbf{\Gamma}$ represent the task-specific head parameters for a pretext task and $\mathcal{L}_{SSL,i}$ represent the loss for $i^{th}$ graph. Then, the total loss for $N$

graphs can be expressed as:

$$\mathcal{L}_{USSL} = \sum_{i=1}^{N} \mathcal{L}_{SSL,i} \left( \mathbf{X_i}, \mathbf{A_i}; \mathbf{\Theta_i}, \mathbf{\Phi}, \mathbf{\Gamma} \right). \tag{8}$$

The total loss $\mathcal{L}_{USSL}$ is used to simultaneously learn the parameters $\{\mathbf{\Theta_i}\}$, $\mathbf{\Phi}$ and $\mathbf{\Gamma}$ in an end-to end manner. The U-SSL loss can also be generalised to any number of tasks, which is presented in Appendix 8.2. At the time of downstream task learning, new heads are appended to the model, parameterized in $\mathbf{\Psi_i}$, which are learnt separately for each graph by keeping the learnt parameters $\{\mathbf{\Theta_i}\}$, and $\mathbf{\Phi}$ unchanged. The pre-training and fine-tuning of U-SSL models are depicted in Fig. 1(b) and 1(c), respectively.

### 3.5 GRAPH TRANSFORMER-BASED U-SSL MODEL

In this work, we construct a U-SSL model that learns universal representations with citation network datasets. We use linear layers as graph-specific encoders, and a powerful graph transformer, i.e., neighbourhood aggregation graph transformer (NAGphormer) (Chen et al., 2023a) as the URL module. NAGphormer is a powerful SOTA graph transformer that leverages position embedding and local neighbourhood-based feature aggregation to tokenize nodes in a graph (Chen et al., 2023a). This approach, referred to a Hop2Token allows encoding both local information and position information in the tokens, which are then used by a tandem of transformer blocks to learn node classification tasks. The NAGphormer has been shown to unanimously outperform SOTA GNNs and GTs for node classification on small-scale and large-scale graphs (Chen et al., 2023a). We construct the URL module of our U-SSL model with multiple transformer encoder layers, followed by an attention-based readout layer employed in NAGphormer. However, unlike NAGphormer, we do not use a multi-layer perceptron module after the attention-based readout, and consider the outputs of the readout as final representations learnt by the model.

## 4 EXPERIMENTS

**Experimental setup** We consider the downstream task of node classification in citation network datasets. We consider 6 benchmark graphs: `CoraFull`, `Cora-ML`, `DBLP`, `Citeseer`, `PubMed` and `OGBN-arxiv` in our study. Our universal model is inspired from NAGphormer, so the URL module (parameterized in $\mathbf{\Phi}$) is constructed with the transformer encoder used in NAGphormer. The URL module has 4 encoder layers with embedding dimension of 256 and 8 attention heads in each layer. The final encoder layer is followed by an attention-based readout, as used in NAGphormer. We employ Laplacian position embedding of the nodes (of size 15) to additionally augment node features with structural information ($F_A$), and obtain the augmented node features of dimension $\tilde{D}_i = D_i + 15$ for graph $\mathcal{G}_i$. This is followed by a linear projection ($F_T$) from the augmented node feature dimension $\tilde{D}_i$ to 256 for graph $\mathcal{G}_i$, which constitute the learnable parameters $\mathbf{\Theta_i}$ of the graph-specific encoders. Although the proposed framework can work with multiple pretext tasks, we consider only one task in this study. Thus, we construct the universal self-supervised model with 1 universal learning module and 1 task-specific module.

The choice of the self-supervision task in our study is guided by the downstream task. Since we are interested in learning features for node classification, we use the pair-wise attribute similarity (**PairSim**) self-supervision task in our study. This task learns an encoder to differentiate between similar and dissimilar nodes, posed as a two-class classification problem. We use one fully connected layer to learn this task. We demonstrate the superiority of the features learnt with U-SSL by evaluating and comparing the performance of models obtained with SSL, U-SSL and supervised learning on node classification for all the graphs. We further train 10 instances of these models for the downstream task to account uncertainty and report the mean and standard deviation of classification accuracy for each experiment. The implementation details are provided in Appendix 8.4.

**Results** We present the advantages of U-SSL over SSL and supervised learning in terms of four aspects: (i) *efficacy*, i.e., improvement in performance compared to SSL, which enables bridging the gap between supervised and self-supervised performance, (ii) *efficiency*, i.e., reduction in training time compared to SSL, (iii) *scalability*, i.e., delivering efficacy and efficiency for larger datasets, and

Table 1: Node classification accuracy of supervised baseline, SSL and U-SSL models. Entries in boldface represent best performance out of SSL and U-SSL. Underlined entries represent U-SSL models that match supervised baseline performance.

| Dataset | Baseline | SSL | U-SSL |
|---------|----------|-----|-------|
| CoraFull | $0.70 \pm 0.007$ | $0.59 \pm 0.003$ | $\mathbf{0.60} \pm 0.003$ |
| Cora-ML | $0.87 \pm 0.004$ | $0.80 \pm 0.002$ | $\mathbf{0.84} \pm 0.001$ |
| DBLP | $0.83 \pm 0.008$ | $0.79 \pm 0.001$ | $\underline{\mathbf{0.83}} \pm 0.001$ |
| Citeseer | $0.94 \pm 0.003$ | $0.83 \pm 0.001$ | $\mathbf{0.86} \pm 0.002$ |
| PubMed | $0.87 \pm 0.008$ | $0.85 \pm 0.002$ | $\underline{\mathbf{0.87}} \pm 0.001$ |

Table 2: Training time per epoch in seconds for self-supervision and downstream task learning.

| Dataset | self-supervision | | Node classification | |
|---------|-----|-------|-------------|---------------------|
| | SSL | U-SSL | Fine-tuning | Supervised baseline |
| CoraFull | 0.294 | ↑ | 0.029 | 0.119 |
| Cora-ML | 0.046 | | 0.004 | 0.020 |
| DBLP | 0.143 | 0.609 | 0.022 | 0.101 |
| Citeseer | 0.046 | | 0.005 | 0.026 |
| Pubmed | 0.134 | ↓ | 0.025 | 0.111 |

(iv) *adaptability*, i.e., the ability to leverage representations learnt through U-SSL on a set of datasets, to learn downstream tasks on new datasets.

*Efficacy:* The node classification accuracy of supervised baseline, SSL and U-SSL models for `CoraFull`, `Cora-ML`, `DBLP`, `Citeseer` and `PubMed` is listed in Table 1. The models obtained with U-SSL outperform the corresponding SSL models, delivering between $1\%$ and $4\%$ improvement in mean accuracy for these datasets. Specifically, U-SSL provides a performance gain of $1\%$ for `CoraFull`, $2\%$ for PubMed, $3\%$ for Citeseer, and $4\%$ for `CoraML` and `DBLP` datasets. We note that `CoraFull` has a large number of classes (70) as well as a large number of nodes $(19,793)$, resulting in a more difficult classification task. Nevertheless, the U-SSL model still produces $1\%$ improvement in accuracy of classification for this dataset. Further, the U-SSL model matches the supervised performance for `DBLP` and `PubMed` datasets, clearly demonstrating the advantage of U-SSL over SSL. These results support our hypothesis, and demonstrate that *there is more to graphs than can be learnt with plain SSL, and learning universal representations across graphs with U-SSL can bridge the gap between supervised and self-supervised performance.* In addition, we note that the total number of parameters for the five SSL models ($\{\boldsymbol{\Theta_i}\}$, $\boldsymbol{\Phi}$) is $14,390,650$, which is 2.46 times $5,831,29$ parameters for the U-SSL model trained with the five datasets. We also observe similar results with co-purchase datasets (see Appendix 8.5). However, including graphs from multiple families does not provide a consistent improvement in performance for all graphs (see Appendix 8.6). This also supports our hypothesis that the underlying similarities between graphs of a family can lead to improvement in performance.

*Efficiency:* We observe that the number of epochs for convergence of SSL and U-SSL models at the time of pre-training are comparable for all datasets (see Appendix 8.7). We therefore report the efficiency in terms of training time per epoch, which are reported in Table 2 for self-supervision and downstream task learning for SSL, U-SSL and supervised models. The time per epoch for building 5 SSL models is 0.663 seconds, which is greater than the time per epoch for building one U-SSL model, i.e., 0.609 seconds for all five datasets. Thus, *U-SSL provides an efficient framework for self-supervised graph representation learning across multiple datasets.*

*Scalability:* We study the scalability of the U-SSL framework to graphs of larger size, specifically `OGBN-arxiv`. We add this dataset to the previous five datasets, and train the model with 6 datasets. The supervised baseline model achieves an accuracy of $0.61 \pm 0.007$, while the SSL model provides an accuracy of $0.46 \pm 0.003$ for the `OGBN-arxiv` dataset. The U-SSL model achieves an accuracy of $0.54 \pm 0.002$, delivering an improvement of $8\%$ in classification accuracy compared to the SSL model. This is a significant gain in performance for a dataset that is much larger than the graphs

Table 3: Ablation results with respect to transformer embedding size. Entries in boldface represent best performance.

| Dataset | Transformer embedding size | | |
|---|---|---|---|
| | 256 | 128 | 64 |
| CoraFull | $\mathbf{0.60} \pm 0.003$ | $0.56 \pm 0.002$ | $0.52 \pm 0.002$ |
| Cora-ML | $\mathbf{0.84} \pm 0.001$ | $0.77 \pm 0.003$ | $0.78 \pm 0.002$ |
| DBLP | $\mathbf{0.83} \pm 0.001$ | $0.81 \pm 0.002$ | $0.80 \pm 0.001$ |
| Citeseer | $\mathbf{0.86} \pm 0.002$ | $0.82 \pm 0.002$ | $0.77 \pm 0.001$ |
| PubMed | $\mathbf{0.87} \pm 0.001$ | $0.85 \pm 0.001$ | $0.84 \pm 0.007$ |

reported in Table 1. This demonstrates that *learning universal representations scales well to graphs of larger size.*

*Adaptability:* Finally, we study the adaptability of the U-SSL framework to new datasets, by examining if the representations learnt from a set of datasets can be used to solve the downstream task for a new dataset. Here, we start with the model obtained with U-SSL of the five previous datasets that has 5 graph-specific modules $\{\mathbf{\Theta_i}\}$, $i \in [1, 5]$. We leverage the modular nature of the U-SSL models, and introduce a new graph-specific module $\mathbf{\Theta_6}$ dedicated to the new graph, keeping the universal representation learning module $\mathbf{\Phi}$ unchanged. We perform self-supervision with the new dataset and learn only $\mathbf{\Theta_6}$, in effect projecting the node features of the new dataset to the universal representation space in a self-supervised manner. We perform this experiment with `OGBN-arxiv` as the new dataset, and find that the adapted model achieves a classification accuracy of $0.538 \pm 0.002$. We observe that this performance is comparable to that of the U-SSL model trained with six datasets discussed the previous section, demonstrating the adaptability of U-SSL models. Thus, one can train a U-SSL model with a set of benchmark datasets, and then simply learn a graph-specific module for a new dataset to achieve comparable performance. This prevents repetitive self-supervised learning with U-SSL for new graphs as they are made available, and is a remarkable feature of the framework that *enables re-use of the learnt representations, thereby reducing the computational cost of building universal models.*

**Ablation** The ablation study of the universal model with respect to the dimension of transformer embedding is reported in Table 3. The performance of the model consistently decreases with smaller embedding dimension for all datasets. The results of ablation with respect to the transformer depth are reported in Table 4. Contrary to Table 3, we observe that the performance of the model does not necessarily increase with greater depth of the URL module. In fact, for all datasets except `CoraFull`, increasing the depth of the URL module from 4 to 6 results in poorer performing model. This suggests that the expressive power, and hence performance of the models is more reliant on having high-dimensional embeddings than a deep URL module.

Finally, we study the ablation of the U-SSL model with respect to the architecture of URL module. We use three GCN, GraphSAGE and GAT layers to construct URL modules, and obtain three different U-SSL models. We compare the performance of the U-SSL models with these architectures and report the results in Table 5. The quantities in parentheses represent the improvement in performance of U-SSL models with respect to SSL models. We observe that the GCN model does not provide any improvement in accuracy for four out of five datasets, and provides an improvement of $3\%$ for `PubMed`. On the other hand, GraphSAGE provides improvements of $1\%$ each for `CoraML` and `Citeseer` datasets, while exhibiting $2\%$ fall in performance for `DBLP`. The NAGphormer-based U-SSL model provides consistent improvement in performance for all datasets, and also outperforms the GNN-based models for majority of the datasets. Thus, the transformer-based U-SSL model provides a better modeling approach to learn universal representations across graphs.

## 5 PERSPECTIVE

**Limitations and future work** The current work demonstrates the advantage of U-SSL by incorporating multiple graphs during self-supervision. However, only one pretext task, i.e., pairwise attribute similarity and one downstream task, i.e., node classification have been studied. It has been shown that including multiple pretext tasks can boost downstream performance Jin et al. (2022). We present

Table 4: Ablation results with respect to transformer depth. Entries in boldface represent best performance.

| Dataset | Transformer depth | | |
|---|---|---|---|
| | 2 | 4 | 6 |
| CoraFull | $0.61 \pm 0.002$ | $0.60 \pm 0.003$ | $\mathbf{0.77} \pm 0.001$ |
| Cora-ML | $0.83 \pm 0.003$ | $\mathbf{0.84} \pm 0.001$ | $0.82 \pm 0.003$ |
| DBLP | $\mathbf{0.83} \pm 0.002$ | $\mathbf{0.83} \pm 0.001$ | $0.80 \pm 0.003$ |
| Citeseer | $0.85 \pm 0.003$ | $\mathbf{0.86} \pm 0.002$ | $0.82 \pm 0.002$ |
| PubMed | $0.86 \pm 0.001$ | $\mathbf{0.87} \pm 0.001$ | $0.85 \pm 0.001$ |

Table 5: Ablation results with respect to architecture of universal representation learning module. Entries in parentheses represent the improvement compared to SSL models. Entries in boldface represent best performance.

| Dataset | URL architecture | | | |
|---|---|---|---|---|
| | NAGphormer | GCN | GraphSAGE | GAT |
| CoraFull | $\mathbf{0.60} \pm 0.003(0.01)$ | $0.60 \pm 0.004(-0.006)$ | $0.55 \pm 0.004(-0.001)$ | $0.45 \pm 0.003(-0.161)$ |
| Cora-ML | $0.84 \pm 0.001(0.04)$ | $\mathbf{0.86} \pm 0.002(-0.005)$ | $0.82 \pm 0.004(0.01)$ | $0.74 \pm 0.002(-0.107)$ |
| DBLP | $\mathbf{0.83} \pm 0.001(0.04)$ | $0.80 \pm 0.002(-0.008)$ | $0.81 \pm 0.002(-0.02)$ | $0.79 \pm 0.002(-0.034)$ |
| Citeseer | $\mathbf{0.86} \pm 0.002(0.03)$ | $0.85 \pm 0.002(0.0001)$ | $0.84 \pm 0.002(0.01)$ | $0.81 \pm 0.003(-0.039)$ |
| PubMed | $\mathbf{0.87} \pm 0.001(0.02)$ | $0.86 \pm 0.002(0.03)$ | $0.83 \pm 0.002(-0.005)$ | $0.84 \pm 0.002(-0.006)$ |

preliminary results of performing U-SSL with two pretext tasks and node classification downstream task in Appendix 8.8, and plan to conduct a more exhaustive study with multiple pretext and downstream tasks, e.g., link prediction and graph classification in the future. The impact of different types of position embeddings and transformer architectures can further be investigated to identify specific embedding strategies that boost performance. The datasets considered here are homogeneous graphs, and the proposed framework can further be extended to heterogeneous graphs to investigate the generalizability of the approach. Finally, the current framework unifies representation learning across graphs, but still needs multiple pretext task heads during pre-training. Future work can be directed to address this, and unify learning across tasks via sequencing output predictions.

**Broader impact** Current research in representation learning is advancing the field towards artificial general intelligence, with foundation models and multi-modal training being major developments in this direction. These models learn representations from different types of data sources, e.g., images, videos and text, that are generalizable across multiple datasets, and at times, across multiple tasks. This work is aligned along these lines, and proposes a framework to build graph foundation models, and learn universal features from multiple graphs.

## 6 CONCLUSION

This work studies the problem of learning universal features across graphs of a family through self-supervision. We present a novel universal SSL framework that constructs foundation model with multiple graph-specific encoders and one universal representation learning module. Specifically, we employ graph-specific encoders to homogenize disparate features from multiple graphs, and the universal module to learn generic representations from the homogenized features. We construct one U-SSL model with a state-of-the-art graph transformer, and with extensive experiments, show that the proposed framework provides an efficacious, efficient, scalable and adaptable approach to learn universal representations from graphs.

## 7 REPRODUCIBILITY STATEMENT

The authors plan to make their GitHub repository public after the review process to ensure reproducibility of the results.

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

## 8 APPENDIX

### 8.1 RELATION TO CONTINUAL GRAPH LEARNING

Continuous learning and graph lifelong learning address the challenge of learning from graph-structured data when the graph structure can evolve (e.g., grow) over time, and/or features of the graph are initially unavailable, and become available over time Liu et al. (2023); Yuan et al. (2023). These paradigms are also used when the task can change over time. However, we propose a framework for self-supervised learning, wherein the graph structure, features and tasks are known ahead of time. We advance the SOTA in SSL by allowing a model to learn simultaneously from multiple datasets - at once, and do not investigate learning as a function of time. Although the ability of our framework to include more graphs as they are made available over time, can be viewed as enabling lifelong learning, the challenges and approaches to lifelong learning focus on one graph growing over time as opposed to multiple graphs being incorporated into one model.

## 8.2 GENERALISATION OF U-SSL LOSS TO MULTIPLE PRETEXT TASKS

The U-SSL loss described in Equation 8 allows pre-training with only one pretext task parameterised in $\mathbf{\Gamma}$. However, the U-SSL framework can be generalised to an arbitrary number $M$ of pretext tasks by modifying the loss function as follows:

$$\mathcal{L}_{USSL} = \sum_{i=1}^{N} \sum_{j=1}^{M} W_j \mathcal{L}_{SSL,i,j} \left( \mathbf{X_i}, \mathbf{A_i}; \mathbf{\Theta_i}, \mathbf{\Phi}, \mathbf{\Gamma_j} \right). \tag{9}$$

Here, $W_j$ represents the weightage provided to the $j^{th}$ pretext task. Note that each task-specific head $\mathbf{\Gamma_j}$ is shared by all the $N$ graphs, and thus, the number of parameters increases only with the number of tasks and is independent of the number of graphs. This feature of U-SSL further reduces the parameter count of a pre-trained model for multiple graphs compared to SSL.

## 8.3 PAIRSIM FOR PRETRAINING

The implementation of our pretext task, i.e., pair-wise node attribute similarity is based on the implementation of Jin et al. (2020) and Jin et al. (2022). First, sets of node pairs with the highest similarity and dissimlarity are created as follows:

$$\tau_{sim} = \{(v_i, v_j) | s_{ij} \in \text{top-K of } \{s_{ik}\}_{k=1}^{N}\} \setminus s_{ii}\}, \tag{10}$$
$$\tau_{dissim} = \{(v_i, v_j) | s_{ij} \in \text{bottom-K of } \{s_{ik}\}_{k=1}^{N}\} \setminus s_{ii}\}. \tag{11}$$

Here, $s_{ij}$ represents the similarity between node features of $v_i$ and $v_j$. This is achieved in out implementation by creating a K-neighbours graph from the node features. The loss function for training can then be expressed as:

$$\mathcal{L}_{PairSim} = \sum_{(v_i, v_j) \in \tau} NLL(f(|h_i - h_j|), y_{ij}), \tag{12}$$

with $\tau = \tau_{sim} \bigcup \tau_{dissim}$ and $h_i$ is the embedding of the $i^{th}$ node, $f$ being the pre-training head and $y_{ij}$ being the target, such that $y_{ij} = 1$ for positive node pair, i.e., $(v_i, v_j) \in \tau_{sim}$ and $y_{ij} = 0$ for negative node pair, i.e., $(v_i, v_j) \notin \tau_{dissim}$.

## 8.4 IMPLEMENTATION DETAILS

All experiments are performed on an NVIDIA DGX-A100 Workstation with four A100 GPUs, each with 40 GB memory. Software is implemented using PyTorch Geometric software library. The implementation of pair-wise attribute similarity is adapted from the implementation of Jin et al. (2022). The official implementation of NAGphormer Chen et al. (2023a) is used to construct the URL module of all models. The Adam optimizer is used to learn the parameters of all models. The base learning rate is set to $1e^{-3}$ for pre-training and supervised learning, and $1e^{-2}$ for fine-tuning of SSL and U-SSL models. A learning rate scheduler that reduces the learning rate when the loss does not decrease for 50 epochs is employed. Self-supervision is performed for 2500 epochs, and fine-tuning is performed for 1000 epochs for SSL and U-SSL models. Supervised baseline models are trained for 500 epochs.

## 8.5 UNIVERSAL SELF-SUPERVISION WITH CO-PURCHASE NETWORKS

We compare the performance of SSL, U-SSL and supervised baselines for the co-purchase family of graphs with `computers`, `photo` datasets. The downstream node classification performance for the three models are shown in Table 6. The results are consistent with those observed for the citation datasets. We obtain 2% improvement for `computers`, and 1% improvement for `photo`. This adds to the results discussed in Section 6, and shows that U-SSL can learn generalisable features for diverse families of graphs.

## 8.6 IMPACT OF GRAPHS FROM MULTIPLE FAMILIES

In the previous results, we consider graphs belonging to one family (citation networks or co-purchase networks) and show that U-SSL learns better features than SSL. We also investigate if including

Table 6: Node classification accuracy of supervised baseline, SSL and U-SSL models for co-purchase datasets. Entries in boldface represent best performance out of SSL and U-SSL.

| Dataset | Baseline | SSL | U-SSL |
|---------|----------|-----|-------|
| computers | $0.90 \pm 0.007$ | $0.83 \pm 0.001$ | $\mathbf{0.86} \pm 0.001$ |
| photo | $0.94 \pm 0.004$ | $0.91 \pm 0.001$ | $\mathbf{0.92} \pm 0.001$ |

Table 7: Node classification accuracy of supervised baseline, SSL and U-SSL models for citation and co-purchase datasets. Entries in boldface represent best performance out of SSL and U-SSL. Underlined entries represent U-SSL models that match supervised baseline performance.

| Dataset | U-SSL (2 families) | U-SSL (1 family) |
|---------|--------------------|--------------------|
| CoraFull | $0.60 \pm 0.003$ | $\mathbf{0.60} \pm 0.003$ |
| Cora-ML | $\mathbf{0.85} \pm 0.002$ | $0.84 \pm 0.001$ |
| DBLP | $0.81 \pm 0.001$ | $\underline{\mathbf{0.83}} \pm 0.001$ |
| Citeseer | $0.85 \pm 0.004$ | $\mathbf{0.86} \pm 0.002$ |
| PubMed | $0.86 \pm 0.002$ | $\underline{\mathbf{0.87}} \pm 0.001$ |
| computers | $0.85 \pm 0.001$ | $\mathbf{0.86} \pm 0.001$ |
| photo | $\underline{\mathbf{0.92}} \pm 0.001$ | $0.92 \pm 0.001$ |

graphs from more than one family also results in better performance. To achieve this, we perform combined training with all the 5 citation networks and 2 co-purchase networks, and summarise the results in Table 7. We observe that out of the 7 datasets, the performance of U-SSL is better (in comparison to SSL) for 1 dataset, worse for 4 datasets, and unchanged for 2 dataset. Based on these results, we cannot claim that U-SSL can always learn better representations when trained across multiple families of graphs. This result corroborates the reasoning behind our hypothesis, i.e., graphs of the same family exhibit commonalities, and thus a combined learning framework can leverage the underlying common patterns to improve the performance.

## 8.7 CONVERGENCE OF SSL AND U-SSL MODELS FOR CITATION NETWORKS

The training convergence of self-supervision for the citation networks is shown in Figure 2. We can see that the convergence of all the models (SSL and U-SSL) are similar, which allows us to compare their efficiency with the training time per epoch, as presented in Table 2.

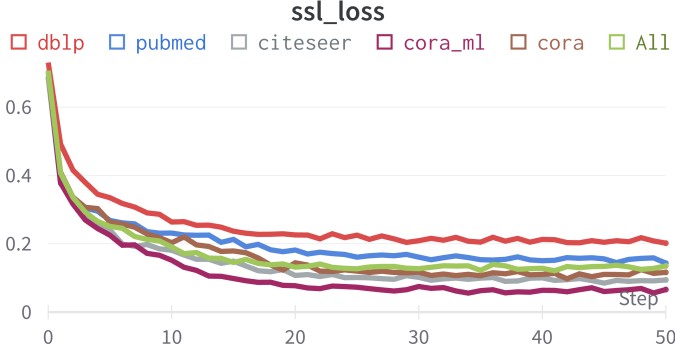

Figure 2: Convergence of pre-training of SSL and U-SSL models. The training loss is logged for every 50 epochs. All models converge with similar rates, allowing us to compare with time per epoch.

Table 8: Node classification accuracy of supervised baseline, SSL and U-SSL models for citation datasets, pretrained on one and two pretext tasks. Entries in boldface represent best performance.

| Dataset | U-SSL (1 task) | U-SSL (2 tasks) |
|---|---|---|
| CoraFull | 0.60 | **0.66** |
| Cora-ML | **0.84** | 0.81 |
| DBLP | **0.83** | 0.81 |
| Citeseer | 0.86 | **0.88** |
| PubMed | **0.87** | 0.86 |

## 8.8 UNIVERSAL SELF-SUPERVISION WITH TWO PRETEXT TASKS

In addition to using the pair-wise node attribute similarity, we also use the pair-wise node distance as an additional pretext task to perform self-supervision. We consider the five citation datasets, and construct the U-SSL model with five graph-specific modules, one universal representation learning module and two task-specific heads for pre-training. We use the loss function described in Equation 9 to tune the parameters $\Theta_i$, $\Phi$ and $\Gamma_j$, $\forall\, i \in [1,5]$ and $\forall\, j \in [1,2]$. The node classification accuracy of the models are shown in Table 8. We can see that pre-training with two tasks results in 6% improvement in performance for CoraFull and 2% improvement for Citeseer. It is noteworthy that while performing self-supervised learning with multiple tasks, weighing the loss for each task is typically performed to achieve an improvement in performance. However, we have not performed a search for the optimal weights ($W_j$ in Equation 9), and have assigned equal weights to both the tasks, i.e., $W_1 = W_2 = 1$. Even with this configuration, we obtain an improvement in performance for two datasets. These results support the general effectiveness of our framework in improving the performance of features learnt through self-supervised learning. Future studies will be aimed at improving optimising the weights of different tasks to achieve consistent improvement in performance.

