# OpenReview forum: "There is More to Graphs than Meets the Eye: Learning Universal Features with Self-supervision"
_ICLR.cc/2024/Conference — Submitted to ICLR 2024_

### Official Review · Reviewer_yAZs · 2023-10-30

**Soundness:** 2 fair
**Presentation:** 3 good
**Contribution:** 2 fair
**Rating:** 3
**Confidence:** 4

**Summary:**

This work studies a framework for learning universal representations from multiple graphs through self-supervision, for node classification tasks. The proposed framework consists of graph-specific encoders that homogenize the distinct node features with different sizes and the universal encode that learns generic features from the homogenized features across different graphs. The framework is evaluated on node classification tasks with benchmark citation graphs, and the authors show the performance and efficiency improvements of the proposed method (namely U-SSL) over the self-supervised learning on individual graphs.

**Strengths:**

* This paper addresses the important and novel problem of learning universal representations across different graphs for node classification tasks.
* The proposed framework, consisting of graph-specific and universal encoders, is intuitive and easy to understand/implement.
* This paper is generally well-written.

**Weaknesses:**

There are some weaknesses in the experimental setups and results, as follows:
* The authors consider only the citation graphs to validate the effectiveness of the proposed universal representation learning framework. In this vein, the proposed method may not be generalizable to more complex networks, such as social networks or code graphs.
* For the citation graphs that are mainly considered in this paper, the authors may not have to use graph-specific encoders to homogenize the features from different graphs. In particular, for citation graphs, we can use the abstract of each paper to generate initial node features, and subsequently we can use the shared vocabulary or a certain method that can encode abstracts in a unified manner (e.g., using LMs to embed them), which means the proposed graph-specific encoders may not be worthwhile to use.
* In Table 1 and Table 2, if the performances of the proposed U-SSL are lower than the performances of the baseline full fine-tuning methods, while these baselines are more efficient than the proposed U-SSL, what are the advantages that we can grab from using the proposed U-SSL?

Also, there is a weakness in the generability of the proposed method, as follows:
* The proposed method seems applicable to only the graphs with features, while real-world graphs sometimes do not have initial node features.

**Questions:**

* Do you use the same backbone model for baseline, SSL, and U-SSL methods?
* When reporting the main results (Table 1), I am wondering why not include the OGBN-arxiv dataset during pre-training, and rather show the results with pre-training on all datasets including the OGBN-arxiv dataset as the extra. I assume that if similar citation networks are used more during pre-training of the proposed U-SSL, the downstream performance may be further increased.

---

> ### Author Response · Authors · 2023-11-13
> **Clarification on backbone model and learning without features**
>
> Thank you for your comments and suggestions. We appreciate your feedback on the paper. We provide an explanation of some of your comments here. We are conducting more experiments to address your other comments and will update soon.
>
> **Graphs without features**:
>
> The current framework relies on node features to learn embeddings. However, in case of graphs that do not have pre-defined features, we can use, for example, hand-crafted features such as degree or q-hop shortest path distance [1] as node features to train a model.
>
> **Backbone model**:
>
> We use the same backbone model for the baseline, SSL and U-SSL methods.
>
> [1] Jiao, Q., Jin, Y., & Liu, Y. (2021, December). Node classification without features using graph convolutional network. In 2021 IEEE 2nd International Conference on Information Technology, Big Data and Artificial Intelligence (ICIBA) (Vol. 2, pp. 991-994). IEEE.

---

> ### Comment · Reviewer_yAZs · 2023-11-22
>
> Thank you for your response. I have read it and I still have concerns about experiments. In particular, the authors only evaluate their model, namely U-SSL, on citation benchmark datasets but also the proposed U-SSL (self-supervised learning framework) does not have advantages over the full-finetuning, in terms of performance gains. Therefore, I maintain my score.

---

### Official Review · Reviewer_Z55K · 2023-10-31

**Soundness:** 2 fair
**Presentation:** 2 fair
**Contribution:** 2 fair
**Rating:** 3
**Confidence:** 4

**Summary:**

This paper introduces a novel concept and explores a new problem domain - Universal self-supervised learning on graphs. In the case of other modalities such as images and natural language, models pre-trained on large datasets tend to generalize well to other datasets. However, achieving similar generalizations for graph data is notoriously challenging. This paper aims to address this challenge through universal self-supervisions, which involve training an Encoder model on various datasets and pretext tasks to obtain universal embeddings. Empirical results demonstrate that the proposed approach bridges the gap between self-supervised and supervised performance.

**Strengths:**

- The research problem is intriguing and holds practical significance.
- The authors provide a comprehensive definition of the concepts used in this paper, such as 'universal self-supervision'.
- The proposed framework is straightforward and easy to comprehend.

**Weaknesses:**

- Concerns about graph-specific encoders: Regarding the alignment of feature dimensions across different graph datasets using a graph-specific encoder, it raises the question of how the proposed method generalizes to new, unseen datasets when there is no learned graph-specific encoder available. Since the aim of this paper is to learn universal node embeddings, it would be interesting to explore how the approach adapts to new datasets where a graph-specific encoder has not been learned.

**Concerns about the experiments**:

- The experimental setup used in this paper appears to be unconventional. Firstly, the authors explicitly choose NAGhormer as the encoder, which avoids direct comparisons with traditional self-supervised methods based on message-passing GNNs. However, NAGhormer does not seem to be the primary contribution of this paper. I recommend that the authors consider conducting experiments with a GCN-based encoder and compare their approach with standard graph-specific self-supervised models, such as contrastive methods.
- Furthermore, in the experimental section, the authors claim to use PairSim as a self-supervised task. While I am well-acquainted with self-supervised learning on graphs, such as contrastive methods and graph autoencoders, I am not familiar with PairSim. It is crucial that the authors provide a detailed explanation of the self-supervised task they use, preferably in the form of an objective function. They should also clarify why they chose this particular task, and if a different task (such as contrastive learning) were adopted, what impact it would have on the experimental results.
- Lastly, the experimental results provided by the authors are not particularly convincing. The results in Table 1 appear to be rather perplexing, as the authors did not clarify the dataset splits, which, to my knowledge, do not align with the common splits used for the corresponding baselines. Furthermore, the authors claim that their proposed method narrows the performance gap between self-supervised and supervised models, but the experimental results do not seem to support this claim. I would suggest that the authors reconsider conducting experiments under more widely adopted self-supervised settings, such as those used in contrastive learning and masked graph autoencoder methods.

**Questions:**

Please see the Weaknesses part.

---

> ### Author Response · Authors · 2023-11-13
> **Clarification of adaptation and experiments**
>
> Thank you for your comments and suggestions. We appreciate your feedback on our work. We provide our explanation on some of your comments.
>
> **Graph-specific encoders**:
>
> The proposed approach can generalise to unseen datasets by learning a new graph-specific encoder. For instance, if we have a U-SSL model trained on 5 datasets, and we want to adapt the model to a new dataset, then it can be achieved by learning a 6th graph-specific encoder, keeping the universal representation learning module unchanged. We have demonstrated this with `ogbn-arxiv` as the new dataset and adapting a U-SSL model trained on 5 citation datasets by learning the new graph-specific module. Please see Page 8 *Adaptability* of the manuscript.
>
> In the current framework, we cannot think of a way to propose a method to adapt the model to a new dataset without learning the new graph-specific encoder, since there is no other way to transfer the node features from their original space to a shared space for the universal module to process.
>
> **Experiments**:
>
> ***Message passing-based GNNs***: We have conducted the study with two GNN models - GCN, GraphSAGE - and reported our findings in Table 5 of the manuscript. We will also add results of an attention-based GNN in the paper soon.
>
> ***SSL Task***: We will provide the detailed description of PairSim in the revised version of the manuscript. PairSim was used in [1] as a pretext task that provides the best performance for downstream node classification task. This task was also adopted in [2] to automate graph self-supervised learning with multiple tasks.
>
> ***Results in Table 1***:
>   - *Data splits*: In the pre-training stage, positive and negative samples are drawn randomly, and there is no train-val-test split. This is because we are interested in learning the features and not worried about overfitting in this stage. This is also in line with the implementation of [2]. In the fine-tuning stage for node classification, we use the `train_mask` attribute of the graph datasets in PyTorch Geometric.
>   - *Narrowing the gap between SSL and supervised performance*: Table 1 shows that the performance of U-SSL is always better than that of SSL, which validates our claim about the gap. We also show that the U-SSL model achieves supervised baseline performance for two graphs, thereby **closing the gap between SSL and supervised baseline**.
>
>
> We are conducting more experiments to address your other concerns, and will update soon.
>
> [1] Jin, W., Derr, T., Liu, H., Wang, Y., Wang, S., Liu, Z., & Tang, J. (2020). Self-supervised learning on graphs: Deep insights and new direction. arXiv preprint arXiv:2006.10141.
>
> [2] Jin, W., Liu, X., Zhao, X., Ma, Y., Shah, N., & Tang, J. (2021, October). Automated Self-Supervised Learning for Graphs. In International Conference on Learning Representations.

---

> > ### Comment · Reviewer_Z55K · 2023-11-17
> > **Wha't the detailed split ratio?**
> >
> > Thank you for your response. In your reply, you mentioned that the split comes from the PyG library. Could you specify the exact split ratio used? As far as I know, and as mentioned by Reviewer w7Bu, PyG provides the official split, so the performance data provided for the node classification task seems quite unusual. It gives the impression that the authors may not be familiar with these datasets.

---

> > > ### Author Response · Authors · 2023-11-23
> > > **Detailed split ratio and performance**
> > >
> > > The exact train-test split ratio for all the datasets is the default from PyTorch Geometric, which is: 20 samples per class for training and 1000 samples for testing.
> > >
> > > In addition, the performance values that are reported in the manuscript are, however, different from those observed in the related literature because of the different architecture adopted in our work. We adopt a transformer that learns in an inductive manner, while the most common GCN performs transductive learning.

---

### Official Review · Reviewer_sutU · 2023-11-01

**Soundness:** 3 good
**Presentation:** 3 good
**Contribution:** 1 poor
**Rating:** 3
**Confidence:** 4

**Summary:**

The paper proposes to learn universal feature representations to facilitate cross-graph self-supervised learning. The model consists of two modules: the graph-specific module projects node features of different graphs to the same space; and then the universal representation learning module gets the features as input and share parameters for graph learning. The authors claim it can be a good way to train foundation graph models.

**Strengths:**

(1) The paper presents a very simple concept which tends to contribute more on the engineering side but useful in practice.
(2) The writing and organization is generally clear.
(3)  Comparatively comprehensive experiments on efficacy, efficiency and scalability

**Weaknesses:**

1. The major concern is about the novelty. The method is too straightforward and we do not see enough insights here.
2. There are some ambiguity about the setting of universal multi-graph SSL. For the pre-training tasks on molecule graphs such as Rong et al. (2020), they are also training on multiple graphs. However the paper actually differs from these papers since it implicitly assumes that the have graphs with node attributes from different domains or with different dimensions. It is generally suitable for those big graphs with node-level/edge-level downstream tasks. But the authors do not make this point very clear.
3. In the experiments, the choice of the self-supervision task in our study is guided by the downstream task. That also looks too heuristic or too specific, because in many cases pretraining is used to train a foundation model which can be generalizable to many different tasks.

**Questions:**

Please refer to weakness.

---

> ### Author Response · Authors · 2023-11-13
> **Clarification on novelty and insights from experiments**
>
> Thank you for your comments and suggestions. We appreciate your feedback on our work. We provide clarifications on the novelty of the work and insights obtained from the experiments.
>
> **Novelty**:
>
> The novelty of the article lies in the unified pipeline that allows us to learn features from multiple graphs at the same time, in an inductive manner, and with shared parameters that leads to a more compact model. Dataset alignment is an important challenge in foundation models. In vision, this is achieved by resizing images of different sizes. In language, this is achieved by using a large common vocabulary. However, alignment of multiple graphs with disparate features is non-trivial. To address this challenge, we adopt an elegant approach and introduce learnable graph-specific encoders and train the model end-to-end such that the individual graph-specific as well as universal features are learnt simultaneously.
>
> **Insights**:
>
> We find the following insights from the experiments:
>
> 1. Training with multiple graphs of a family leads to better node embeddings and thus, better downstream performance. This is because graphs of a family share commonalities that can be leveraged to learn universal features.
> 2. Training with multiple graphs from different families does not lead to consistent improvement in performance. This is because graphs of different families can exhibit very different patterns that limit the universality of features learnt through simultaneous training.
> 3. The U-SSL model allows adaptability and can accommodate newly available graphs by learning only the graph-specific module, i.e., learning only the alignment-related parameters, and leveraging universal parameters learnt during pre-training. This has practical significance on the amount of computational effort needed to train foundation models.
> 4. Training on multiple tasks does not provide consistent improvement in performance, and further research is needed to appropriately combine the objective functions from the tasks, as conducted in [1] to improve the performance.
>
>
> We are working towards addressing the rest of your comments, and will update soon.
>
> [1] Jin, W., Liu, X., Zhao, X., Ma, Y., Shah, N., & Tang, J. (2021, October). Automated Self-Supervised Learning for Graphs. In International Conference on Learning Representations.

---

### Official Review · Reviewer_w7Bu · 2023-11-02

**Soundness:** 2 fair
**Presentation:** 3 good
**Contribution:** 2 fair
**Rating:** 5
**Confidence:** 4

**Summary:**

The manuscript proposes a generic framework designed for universal representation learning on graphs with different input features. By leveraging multiple graphs of a family, the proposed method can improve the quality of
learned representations, similar to multi-task learning. Experiments show that the method outperforms traditional SSL counterparts and obtains comparable results with supervised learning on some datasets.

**Strengths:**

1. Overall, the paper is well-written and easy to follow.
2. The paper studies an interesting research question of how to utilize knowledge contained in different graphs. As we know, graph datasets usually come with different node/edge features, which significantly hinders the study of transfer learning and foundation models. The authors adopt a simple approach to map the raw features from different domains into one shared feature space, and then train a shared graph encoder on top of the universal space. The pipeline makes sense to me and shows some promising results.
3. The framework seems to show some transferability. For example, when trained on [CoraFull, Cora-ML, DBLP, Citeseer, PubMed] and adapted on Arxiv, the result is better than performing SSL only on Arxiv itself. This demonstrates that training a shared graph encoder given a universal feature space can indeed help transfer learning.
4. In section 8.5, the authors show that including graphs with a larger domain gap negatively affects the quality of the universal embeddings. This finding is consistent with my understanding and it'll be better if the authors conduct more experiments to further demonstrate it.

**Weaknesses:**

1. The authors only implement the framework with one specific graph encoder, i.e., NAGphormer. The results will be much more convincing with more backbones included.
2. Similarly, they only use one pretext objective for SSL and test with one downstream task, i.e., SSNC.
3. The experiments lack details, and no code is available.

**Questions:**

1. What's the data split used in the experiments? I looked everywhere but found nothing. The results for node classification seem weird to me, e.g., even a simple 2-layer GCN can achieve over 70% accuracy on Arxiv, yet the supervised baseline in the manuscript is only 61%. The authors must explain the gap between the results reported in the manuscript and the ones commonly found in related literature.
2. How to do the self-supervised learning (PairSim)? How to define the 'similar' or 'dissimilar' nodes without label information? Please add the equation of the training objective to the revised version.
3. The results in Table 1 are strange. In conventional graph-ssl literature, e.g., DGI[1], the performance of supervised GCN and SSL variants is close. However, in Table 1, there is a significant gap between the supervised baseline and SSL methods for some datasets.
4. Why add an additional data augmentation step before feeding into the linear layers? The authors should conduct an ablation study to show the effect of such graph-aware augmentation.
5. If the authors want to claim transferability (so-called Adaptability in the manuscript) of their framework, more experimental results and analysis should be included. Only the result on Arxiv is not convincing enough.
Overall, this paper is interesting in its research question, and the overall framework also makes sense to me. However, many technical details are missing and reproducibility is limited.

Reference:
[1] Veličković, Petar, et al. "Deep graph infomax."

---

> ### Author Response · Authors · 2023-11-13
> **Explanation of experiment details, dataset split and graph-aware data augmentation**
>
> Thank you for your constructive comments and suggestions. We appreciate your positive feedback. We are providing our partial response here, and will update with results from new experiments soon. We hope our responses below clarify some of your questions.
>
> **Experiment details and reproducibility**:
> - Our implementation of PairSim is derivative of [1], who implement a host of SSL tasks for graphs including pairwise similarity, pairwise distance, etc. The following steps are used to generate labels for PairSim:
>
>   1. A k-neighbours graph is constructed based on the node features of the graph `x` and predefined value of `k` (10 in our experiments).
>   2. Positive and negative node pairs are sampled from the graph based on k-neighbours graph obtained in step 1.
>   3. The difference in embeddings (i.e., the output of the universal backbone) of node pairs are fed to a predictor (linear layer in our case). The NLL loss is used to train the model with the target as 1 for positive pairs and 0 for negative pairs.
> - We will upload our latest U-SSL code as a .zip file in the Supplementary material soon.
>
> **Data split**:
> - In the pre-training stage, positive and negative samples are drawn randomly, and there is no train-val-test split. This is because we are interested in learning the features and not worried about overfitting in this stage. This is also in line with the implementation of [1]. *Note: For a graph with 100 nodes, one has `2^100` node pairs (positive and negative included), of which we train on `num_samples X num_epoch=8000*300` pairs.*
> - In the fine-tuning stage for node classification, we use the `train_mask` attribute of the graph datasets in PyTorch Geometric.
>
> **Additional data augmentation step**:
> We do not have any extra data augmentation step before feeding into the linear layers. We employ Hop2Token, which is a part of the NAGphormer pipeline, and do not introduce any additional pre-processing or post-processing steps in our framework. This step is an integral part of NAGphormer without which, we cannot process the data, so it is not possible to conduct an ablation with respect to Hop2Token.
>
> We are working towards addressing your other comments, and conducting more experiments, and will update soon.
>
> [1] Jin, W., Liu, X., Zhao, X., Ma, Y., Shah, N., & Tang, J. (2021, October). Automated Self-Supervised Learning for Graphs. In International Conference on Learning Representations.

---

> > ### Comment · Reviewer_w7Bu · 2023-11-21
> > **Response to author rebuttal**
> >
> > Thank you to the authors for your response.
> >
> > However, the response did not address my concerns regarding the experimental settings and reproducibility. Meanwhile, the overall contribution of the work falls short of the bar of ICLR, so I will maintain my initial score.

---

> ### Author Response · Authors · 2023-11-22
> **Reproducibility and explanation of results**
>
> Thank you for your patience and for your comments.
>
> **Reproducibility**: We have uploaded the code as a .zip file in the supplementary material.
>
> **Difference in performance**: The observed difference in performance between the results reported in the paper and other methods in the literature is due to the different architecture adopted. We use a transformer model, which learns in an inductive manner, as opposed to a GCN, which is transductive in nature, and has access to the entire connectivity of the graph.
>
> **Gap between supervised and self-supervised models**: The gap between supervised and self-supervised models also arises from the fact that we employ an inductive transformer, whereas conventional literature, for instance, DGI uses GCN. We performed the experiments with GCN as the backbone, and observed that the difference in performance is much less compared to the transformer model.
>
> The difference in supervised and SSL performance for the citation datasets for different backbones is as follows:
> | Dataset | NAGphormer| 1-layer GCN |
> |---|---|---|
> | Cora | 11 | 5 |
> | Cora-ML | 7 | 2 |
> | Citeseer | 11 | 6 |
> | DBLP | 4 | 0 |
> | Pubmed | 2 | 0 |
>
> This shows that the large gap that we observe is because of the transformer model, and upon using a 1-layer GCN model, we also get very closely performing SSL and supervised models. However, GCN models do not (consistently) perform better with U-SSL. This can partly be attributed to the the fact that message passing-based GNNs can perform poorly in high-data scenarios, while transformer-based models thrive on vast amounts of data [1]. Since a U-SSL model is trained with more than 1 graph, and thus, sees much more data than an SSL model, GNNs might struggle to perform well with U-SSL, while transformers perform better.
>
> [1] Ma, Liheng, et al. "Graph Inductive Biases in Transformers without Message Passing." International Conference on Machine Learning (2023).

---

### Meta-Review · Area_Chair_1b6W · 2023-12-08

**Metareview:**

The paper presents a novel idea in the domain of graph self-supervised learning. However, the methodology and experimental validation face significant challenges. There are concerns about the novelty of the approach, the specificity of the model implementation, and the clarity of the methodological framework. The lack of detailed experiments and the unavailability of code further limit the reproducibility and credibility of the results. Additionally, the paper seems to focus predominantly on citation graphs, raising questions about its generalizability to other types of networks. These issues, combined with the concerns about the paper's lack of deeper insights and methodological clarity, suggest that the manuscript does not meet the acceptance standards for publication at this time.

In light of these concerns, it is recommended that the paper be rejected. The authors may consider addressing these significant issues in future submissions.

**Justification For Why Not Higher Score:**

As said in meta review

**Justification For Why Not Lower Score:**

NA

---

### Decision · Program_Chairs · 2024-01-16

Reject